# Heterogeneous Photocatalysis as a Potent Tool for Organic Synthesis: Cross-Dehydrogenative C–C Coupling of *N*-Heterocycles with Ethers Employing TiO_2_/*N*-Hydroxyphthalimide System under Visible Light

**DOI:** 10.3390/molecules28030934

**Published:** 2023-01-17

**Authors:** Elena R. Lopat’eva, Igor B. Krylov, Oleg O. Segida, Valentina M. Merkulova, Alexey I. Ilovaisky, Alexander O. Terent’ev

**Affiliations:** N. D. Zelinsky Institute of Organic Chemistry, Russian Academy of Sciences, 47 Leninsky Prospekt, 119991 Moscow, Russia

**Keywords:** Minisci reaction, heterogeneous photocatalysis, *N*-hydroxyphthalimide, titanium dioxide, green chemistry, visible light photocatalysis

## Abstract

Despite the obvious advantages of heterogeneous photocatalysts (availability, stability, recyclability, the ease of separation from products and safety) their application in organic synthesis faces serious challenges: generally low efficiency and selectivity compared to homogeneous photocatalytic systems. The development of strategies for improving the catalytic properties of semiconductor materials is the key to their introduction into organic synthesis. In the present work, a hybrid photocatalytic system involving both heterogeneous catalyst (TiO_2_) and homogeneous organocatalyst (*N*-hydroxyphthalimide, NHPI) was proposed for the cross-dehydrogenative C–C coupling of electron-deficient *N*-heterocycles with ethers employing *t*-BuOOH as the terminal oxidant. It should be noted that each of the catalysts is completely ineffective when used separately under visible light in this transformation. The occurrence of visible light absorption upon the interaction of NHPI with the TiO_2_ surface and the generation of reactive phthalimide-*N*-oxyl (PINO) radicals upon irradiation with visible light are considered to be the main factors determining the high catalytic efficiency. The proposed method is suitable for the coupling of π-deficient pyridine, quinoline, pyrazine, and quinoxaline heteroarenes with various non-activated ethers.

## 1. Introduction

Heterogeneous photocatalysis in organic synthesis is a young and fast-growing area [1,2,3,4,5]. The semiconductor materials used in photocatalysis are inexpensive and widely available; their advantages include the ease of separation from organic products, stability and recyclability [1,5]. However, the development of this area is still hindered by several formidable obstacles, such as low catalytic efficiency due to the low degree of charge separation in photoexcited states and the fast recombination of electron–hole pairs [6,7], low visible light absorption and low selectivity due to the strong oxidation power of photogenerated valence-band (VB) holes in popular semiconductors (TiO_2_, ZnO, Bi_2_O_3_, WO_3_, etc.) [1,8]. This situation is reflected in the comparatively low number of synthetic methods in fine organic synthesis based on heterogeneous photocatalytic systems compared to the mainstream applications of heterogeneous photocatalysis: oxidative destruction of pollutants [9,10,11], hydrogen generation [12,13], CO_2_ reduction [14,15,16] and water splitting [17].

Currently, the scope of synthetic transformations enabled by heterogeneous photocatalysis is much less diverse compared to the scope of homogeneous photoredox-catalyzed reactions. Heterogeneous catalysis is mainly used in comparatively simple reactions; for example, alkylarene benzylic oxidation [18,19,20], the oxidation of benzylamines [4,5,21,22], alcohols [4,5] and sulfides [4,23], oxidative esterification [4], nitro-group reduction [4], tiol-ene reaction [24], alkene amination with aqueous ammonia [25] and the decarboxylation of carboxylic acids [26,27,28]. Cross-coupling reactions are much less developed and usually demand transition metal co-catalysts, such as palladium or nickel complexes [29,30,31,32,33].

UV irradiation, which is used frequently for the excitation of heterogeneous photocatalysts, is inconvenient due to safety issues, the comparatively high cost of UV light sources, incompatibility with common laboratory glassware (UV-transparent quartz is necessary) and possible side reactions due to the high energy of the light. The modification of heterogeneous photocatalysts, such as TiO_2_, in order to shift their photoactivity spectrum from UV to visible light [10,34,35,36,37] is the key task for expanding the scope of their applications in organic synthesis, increasing selectivity and making the of use cheap and available light sources for catalyst activation possible. At present, the following modification approaches have been proposed: the immobilization of dyes (organic compounds or metal complexes) on the photocatalyst surface [34,38,39,40,41], doping with metal ions or non-metal elements [42,43], semiconductor coupling [7,44,45,46,47,48,49] and modification with organic molecules bearing hydroxyl or carboxyl groups [34,50,51,52,53,54,55,56], which demonstrate the occurrence of visible light absorption when adsorbed on the surface of a semiconductor.

NHPI/TiO_2_ is one of the efficient catalytic systems activated by visible light based on industrially available substances (Figure 1). The interaction of NHPI with the TiO_2_ surface leads to the occurrence of visible light absorption, resulting in the photogeneration of phthalimide-*N*-oxyl radicals (PINO) [20,22]. In our previous work [20], we demonstrated that the NHPI/TiO_2_ system could be successfully applied to the aerobic oxidation of alkylarenes under visible light irradiation (Figure 1A). The conceptual novelty of this system arises from the conjunction of heterogeneous photocatalysis with homogeneous radical chain organocatalysis. A distinguishing feature of this system is the migration of PINO into the volume of solution, where the PINO/NHPI catalyzed radical chain process, once initiated on the TiO_2_ surface, produces the target product without the need for additional light absorption [20]. Thus, the energy efficiency of photocatalysis is fundamentally improved by combining heterogeneous photocatalysis with homogeneous organocatalysis. In the presence of additional organocatalyst (2,2,6,6-Tetramethylpiperidin-1-yl)oxyl (TEMPO) the effective oxidative homocoupling of benzylamines [22] was achieved previously (Figure 1B).

In the present study, we demonstrate the successful application of the NHPI/TiO_2_ system to a more challenging cross-dehydrogenative C–C coupling process (Figure 1C). In this case, previously reported CH-oxygenation processes [20] should be suppressed, which is a difficult task. In addition, the process of C–O coupling between NHPI-derived PINO radicals and CH-reagents [57,58,59] must be avoided. The oxidative coupling of ethers with π-deficient *N*-heteroaromatic compounds (a Minisci-type reaction) was chosen as a model reaction due to the practical importance for the functionalization of *N*-containing heterocycles with C–C bond formation. Minisci-type reactions [60,61,62,63,64,65,66,67,68] are based on the addition of nucleophilic C-centered radicals to electron-deficient arenes and represent one of the most important methods for the functionalization of such arenes, along with the nucleophilic aromatic substitution of hydrogen [69,70,71], and functionalization via transition-metal-catalyzed C(sp^2^)–H bond activation [72,73,74,75,76]. The products of the Minisci reaction are of great value for medicinal chemistry [61,64]. Thus, the development of new, milder, more efficient methods tolerant to a large number of functional groups based on Minisci chemistry remains a hot research topic.

To date, many photochemical protocols have been developed for the Minisci reaction, both with the use of metal complex photocatalysts [60,77,78,79,80] and organic photocatalysts [81,82,83]. In some specific cases, the Minisci reaction proceeds without a photocatalyst [84,85,86,87]. At the same time, examples of the application of heterogeneous photocatalysis for the Minisci reaction that are attractive from the practical point of view remain rare [88,89,90,91]. In this work, we demonstrate the use of the developed hetero-/homogeneous NHPI/TiO_2_ photocatalytic system for the Minisci reaction between π-deficient heteroarenes (pyridines, quinolines, isoquinolines, pyrazines, and quinoxaline) and non-activated ethers.

## 2. Results and Discussion

### 2.1. Optimization of Photocatalytic System Composition

Based on our previous work [20], TiO_2_ with high specific surface area (anatase nanopowder, Hombikat UV100) and industrially available *N*-hydroxyphthalimide were chosen as the components of the photochemical system. Blue LEDs (455 nm) with an input power of 10 W were used as light sources. In the first step, we optimized the conditions of the photochemical cross-dehydrogenative Minisci reaction between 4-methylquinoline **1a** and tetrahydrofuran **2a** (Table 1). *Tert*-butyl hydroperoxide (TBHP) was used as an inexpensive, easily available and metal-free oxidant.

The starting conditions (10 mg of TiO_2_, 20 mol.% of NHPI, 4 mmol of TBHP, 5 h, run 1) yielded 45% of the product **3aa**. The absence of either TiO_2_ or NHPI resulted in the zero conversion of **1a** (runs 2, 3), proving that both components of the catalytic system are essential. Without *t*-BuOOH, the reaction proceeded with low efficiency: only trace amounts of the product were formed (run 4). As a rule, the addition of a strong Brønsted acid, such as HCl [85] or TFA [77,79,82,84,86], increases the efficiency of the Minisci reaction. Acids protonate π-deficient *N*-containing heterocycles, making them more susceptible to attack by nucleophilic C-centered radicals [67]. However, in our case, the addition of trifluoroacetic acid (TFA, run 5) had no significant effect on the yield and conversion. The addition of 0.5 mL of water resulted in a drop in **3aa** yield (run 6). Water breaks down the stable suspension of TiO_2_ in THF, causing the catalyst particles to aggregate in the water droplets. Both an increase and a decrease in the amount of THF lead to a decrease in the yield of **3aa** (runs 7, 8). The dilution of the reaction mixture with such co-solvents as hexafluoroisopropanol (HFIP, run 9) and acetonitrile (MeCN, run 10) slowed down the reaction, and dilution with dichloroethane (DCE, run 11) led to the complete suppression of the target process. It is known that hydrogen peroxide can be used as the oxidant for the photocatalytic Minisci reaction [85]. However, the change of the oxidant from TBHP to aqueous H_2_O_2_ led to a dramatic drop in the yield (run 12). The lower efficiency of H_2_O_2_ compared to TBHP can be explained by the fact that H_2_O_2_ can not only initiate free-radical reactions but can also be an inhibitor via the formation of HOO• radicals [92,93,94]. The use of other organic peroxides, such as meta-chloroperoxybenzoic acid (m-CPBA, run 13), cumene hydroperoxide (run 14) and dicumyl peroxide (run 15) led to low yields or did not provide the product at all. Dibenzoylperoxide (BzOOBz, run 16) showed a yield comparable to TBHP, but the formation of a large amount of benzoic acid, which is poorly soluble in the system, complicates the isolation of the products and limits the scalability of the procedure. Therefore, TBHP was chosen as the optimal oxidant. The standard version of the Minisci reaction often uses inorganic persulfates as oxidants. In our system, the use of persulfates was less efficient than TBHP, and led to a significant drop in yield with increasing reaction time, presumably due to the overoxidation of the product (runs 17–20). An inert atmosphere did not increase the selectivity of the process (run 21), so we decided to carry out the reaction under air.

In the next step, we optimized the NHPI/TiO_2_/TBHP ratio and irradiation time to achieve the maximum yield of the coupling product **3aa** (Table 2).

Increasing the amount of TiO_2_ increases the yield of **3aa** (runs 1–4). However, when switching from the TiO_2_ loading of 20 mg to 40 mg, the efficiency increased only slightly. Therefore, the TiO_2_ loading of 20 mg was chosen as the optimal amount. Similarly, large loadings of NHPI resulted in an increase in the **3aa** yield (runs 5–8), but the step from 20 to 40 mol.% of NHPI increased the yield of **3aa** slightly, and a slight drop in selectivity was observed. The optimum excess of THBP was 4 mmol per 1 mmol of **1a** (runs 9–11). The reaction proceeded with almost complete conversion in 8 h (run 15). It should be noted that visible-light-active heterogeneous photocatalyst g-C_3_N_4_ was ineffective for the model coupling reaction under the same conditions (run 16). The conditions of experiment 15 were chosen as optimal for further studies of the substrate scope for the developed method.

### 2.2. Application of the Designed Photocatalytic NHPI/TiO_2_ System to the Minisci Reaction

With the optimal conditions in hand (Table 2, run 15), we have synthesized a wide range of coupling products between *N*-heterocycles and ethers. The scope of ethers was explored first (Figure 2). For substrates demonstrating lower conversions compared to **1a**, the reaction time increased in some cases up to 48 h (the reaction times and conversions are given in Figure 2).

Among the tested ethers, we obtained the best result with THF: after 8 h of reaction, the almost complete conversion of 4-methylquinoline **1a** and a high yield of product **3aa** (89%) were observed. As a rule, the reaction proceeds more slowly and with lower selectivity for other ethers. In the reaction of 4-methylquinoline with 2-methyltetrahydrofuran **2b**, a mixture of products **3ab** (as a diastereomeric mixture, major) and **3ab’** (minor) was observed. The observed regioselectivity can be explained by the fact that although the hydrogen atom abstraction is most favored from the weakest tertiary CH-bond (position 2 of 2-methyltetrahydrofuran) [95], the resulting C-centered radical is more stable and sterically hindered than the secondary radical and reacts less efficiently with 4-methylquinoline. For 1,3-dioxolane **2c**, two isomeric products **3ac** and **3ac’** were formed, and the major product **3ac** corresponds to the breaking of the weakest C2-H bond in 1,3-dioxolane. With dioxane and tetrahydropyran, the reaction proceeded more slowly, but with a longer reaction time, its selectivity decreased simultaneously with an increase in conversion. With glyme, the dehydrogenative coupling product was not observed even after 24 h of reaction.

In the case of diethyl ether as a substrate, the reaction under the standard conditions was not effective due to the immiscibility of Et_2_O and H_2_O contained in TBHP (70% aq.), which led to the aggregation of TiO_2_ particles in water droplets and the low conversion of **1a**. The solution to the problem was the use of anhydrous TBHP, prepared before the reaction (See experimental details for Figure 2). The same problem limited the reaction time for the coupling of **1a** with Et_2_O since the water generated during TBHP reduction accumulated in the reaction mixture and made the TiO_2_ suspension unstable.

In the next step, the scope of the electron-deficient *N*-heterocycles was tested (Figure 3).

*N*-heterocycles with electron-donor groups reacted slower compared to substrates with electron-withdrawing groups, but at the same time, higher selectivity was observed (products **3ba**, **3ea** in comparison with **3ca**). The reaction is sensitive to steric hindrance: 2-chloro-5-bromoquinoline **2d** did not yield the target product of **3da**, presumably due to the presence of a bulky Br substituent near the 4th position of the quinoline. Our photochemical system is also applicable to quinoxalines and pyrazines. It is worth noting that the products of **3ga** and **3ha** have not been previously reported (See Appendix A for additional information). In general, the reaction is inefficient for pyridines with no substituents or with electron-donor substituents (pyridine, picolines, lutidine), but good yields have been obtained for pyridines with electron-acceptor substituents, such as pyridine-3-carboxylic acid methyl ester (product **3ia**). 4-Methylquinoline-*N*-oxide reacted with the preservation of the *N*-oxide function (product **3ja**). Good yields have also been obtained in the reaction with isoquinoline (product **3ka**). In the reaction with imidazo [1,2-a]pyridine **2l**, it was only possible to isolate the product of deep oxidation with the destruction of the ring—**3la’**. It should also be noted that the addition of acid (TFA) afforded increased yields in some cases (products **3ba**, **3ca**, **3ea**, **3ga**, **3ha**,**3ja** and **3ka**).

It turned out that carrying out the reaction to complete the conversion of π-deficient arenes in the NHPI/TiO_2_ photochemical system leads to a sharp drop in selectivity for target product **3**. We assumed that product **3** could undergo further oxidation under the reaction conditions. To find out what role the individual components of the system play in oxidation, we performed control experiments in which the pure reaction product **3aa** was placed under standard reaction conditions or irradiated in an inert atmosphere in the absence of NHPI or TBHP (Figure 4).

Under the standard conditions, an 86% conversion of **3aa** was observed in 8 h (Figure 4, **A**). In the absence of TBHP under an air atmosphere, the product is also oxidized (88% conversion, Figure 4, **B**), which suggests that a significant role in the decomposition of the product is played by air as an oxidant. The primary oxidation product was hydroperoxide **3aa’**, which was detected in a mixture of oxidation products by ^13^C NMR and was confirmed by HRMS (See Appendix A). The ^13^C signal with chemical shift typical for geminal alkoxyhydroperoxide fragment was observed [96]. However, carrying out the reaction under an argon atmosphere (Figure 4, **C**) does not completely suppress the oxidation of product **3aa** since TBHP or residual amounts of oxygen can serve as oxidants. The lowest conversion of the product was observed when the reaction was carried out in an argon atmosphere without the addition of NHPI (Figure 4, **D**), implying that NHPI-derived PINO radicals play an important role in **3aa** oxidation.

Based on the collected data, we proposed the following mechanism (Figure 5). Upon irradiation with visible light, PINO radicals are generated from NHPI on the TiO_2_ surface. Simultaneously, the *tert*-butyl hydroperoxide decomposes on the TiO_2_ surface with the formation of *tert*-butoxyl radicals. *Tert*-butoxyl radicals can regenerate PINO by abstracting a hydrogen atom from the NHPI in solution [59]. *Tert*-butoxyl radicals can also generate *tert*-butylperoxy radicals from *t*-BuOOH [97,98]. Either *tert*-butoxy, *tert*-butylperoxy [99,100,101], or PINO radicals [59,95] can abstract a hydrogen atom from the α-CH bond in ether to form C-centered radical **A**. However, considering the fact that no cross-dehydrogenative coupling was observed without the addition of NHPI, the main role in H-atom abstraction is assumed to be played by the PINO radicals. Then, radical **A** undergoes addition to a heteroarene with the formation of the intermediate radical **B**, which is further subjected to HAT with the retrieval of aromaticity.

## 3. Materials and Methods

### 3.1. General

Room temperature (rt) stands for 23–25 °C.

Commercial TiO_2_ Hombikat UV 100 (anatase, specific surface area, BET: 300 m^2^·g^−1^, primary crystal size according to Scherrer <10 nm) was used as is. *N*-hydroxyphthalimide (NHPI, 98%, Acros Organics), 4-methylquinoline (99%, Acros Organics), 2-methylquinoline (97%, Acros Organics), 2-chloroquinoline (99%, Acros Organics), isoquinoline (97%, Acros Organics), quinoxaline (99%, Acros Organics), pyrazine (99+%, Acros Organics), 2-methylpyrazine (99+%, Acros Organics), Methyl nicotinate (99%, Acros Organics), 2-methoxyquinoline, 5-bromo-2-chloroquinoline were used as is from commercial sources. 4-methylquinoline 1-oxide was synthesized according to the literature procedure [102], 2-(4-bromophenyl)imidazo [1,2-a]pyridine was synthesized according to the procedure in the literature [103]. Bulk g-C_3_N_4_ was prepared analogously to previously reported methods [104,105], and the urea was heated in a covered alumina crucible for 4 h at 550 °C (heating rate 5 °C·min−^1^). MeCN was distilled over P_2_O_5_, and Ethers (THF, 2-Methyltetrahydrofuran, 1,3-dioxolane, 1,4-dioxane, tetrahydropyran and diethyl ether, dimethoxyethane, bis(2-methoxyethyl) ether) were distilled over LiAlH_4_. The reaction mixtures were sonicated in an ultrasonic bath (HF-Frequency 35 kHz, ultrasonic nominal power 80 W) before the irradiation.

Experimental details for Table 1

General reaction conditions: 4-methylquinoline **1a** (1 mmol, 143.2 mg), TiO_2_ (10 mg), NHPI (0.2 mmol, 32.6 mg), *t*-BuOOH (70% aq., 4 mmol, 515 mg), THF **2a** (25 mmol, 2 mL) and a magnetic stir bar (6 × 10 mm) were placed in a 50 mL round-bottom flask. The obtained mixture was sonicated for 5 min in an ultrasonic bath, then magnetically stirred (500 rpm) in a thermostated water bath at 25 °C (±1 °C) under irradiation of 10 W blue LED for 5 h under an air atmosphere (closed flask). Then, the solvent was rotary evaporated, and C_2_H_2_Cl_4_ (40–60 mg, 0.4–0.61 mmol) was added as a standard for NMR yield determination. The reaction mixture was centrifuged, and the NMR spectrum was recorded.

Experimental details for Table 2

4-methylquinoline **1a** (1 mmol, 143.2 mg), TiO_2_ Hombikat UV 100 (2.5–40 mg), NHPI (0.05–0.4 mmol, 8.2–65.2 mg), *t*-BuOOH 70% aq. (1–6 mmol, 129–772 mg) and THF **2a** (25 mmol, 2 mL) and a magnetic stir bar (6 × 10 mm) were placed in a 50 mL round-bottom flask. The obtained mixture was sonicated for 5 min in an ultrasonic bath, then magnetically stirred (500 rpm) in a thermostated water bath at 25 °C (±1 °C) under irradiation of 10 W blue LED for 1–16 h under an air atmosphere (closed flask). Then, the solvent was rotary evaporated, C_2_H_2_Cl_4_ (40–60 mg, 0.4–0.61 mmol) was added as a standard for NMR yield determination. The reaction mixture was filtrated through a Celite layer, and the NMR spectrum was recorded.

Experimental details for Figure 2 and Figure 3

Heterocycle **1** (1 mmol), TiO_2_ (20 mg), NHPI (0.2 mmol, 32.6 mg), *t*-BuOOH 70% aq. (4 mmol, 515 mg), CH-reagent **2** (25 mmol) and a magnetic stir bar (6 × 10 mm) were placed in a 50 mL round-bottom flask. The obtained mixture was sonicated for 5 min in an ultrasonic bath, then magnetically stirred (500 rpm) in a thermostated water bath at 25 °C (±1 °C) under irradiation of 10 W blue LED for 8 h under an air atmosphere (closed flask). If needed, another 4 mmol of the reaction *t*-BuOOH was added, and the reaction mixture was irradiated for another 8 h. At the end of the required time, the reaction mixture was poured into 20 mL of water and extracted with 3×15 mL of CH_2_Cl_2_. The combined organic extracts were washed with 2×20 mL of NaHCO_3_ saturated solution. The extracts were dried over MgSO_4,_ and the solvent was evaporated in a vacuum membrane pump. The residue was purified using column chromatography to afford products **3aa**–**3ka**. For the reaction of **1a** with Et_2_O, anhydrous *t*-BuOOH was prepared. *t*-BuOOH 70% aq. (12 mmol, 1545 mg) was extracted with CH_2_Cl_2_ (10 mL). The organic layer was dried over MgSO_4_, and the solvent was rotary evaporated. The obtained anhydrous *t*-BuOOH was used instead of *t*-BuOOH 70% aq. For the longer reaction times, the new portion of anhydrous *t*-BuOOH (4 mmol, 360 mg) was added each 8 h.

Experimental details for Figure 4

4-methyl-2-(tetrahydrofuran-2-yl)quinoline **3aa** (0.5 mmol), TiO_2_ (10 mg), NHPI (0.1 mmol, 16.3 mg), *t*-BuOOH 70% aq. (2 mmol, 257 mg) and a magnetic stir bar (6 × 10 mm) were placed in a 50 mL round-bottom flask. The obtained mixture was sonicated for 5 min in an ultrasonic bath. For the entries of C and D, the flask was vacuumed and then filled with Ar three times. The mixture was magnetically stirred (500 rpm) in a thermostated water bath at 25 °C (±1 °C) under irradiation of 10 W blue LED for 8 h. The conversion of **3aa** was determined by ^1^H NMR in MeCN using C_2_H_2_Cl_4_ as the internal standard.

### 3.2. Characterization Data of the Cross-Dehydrogenative C–C Coupling Products

**4-Methyl-2-(tetrahydrofuran-2-yl)quinoline 3aa** [91] was isolated using column chromatography (Petroleum ether/EtOAc = 2/1) as a colorless viscous liquid (190 mg, 89%). ^1^H NMR (300.13 MHz, CDCl_3_) δ 8.07–7.99 (m, 1H), 7.92–7.85 (m, 1H), 7.66–7.58 (m, 1H), 7.48–7.41 (m, 1H), 7.40 (s, 1H), 5.10 (t, *J* = 6.9 Hz, 1H), 4.16–4.08 (m, 1H), 4.02–3.94 (m, 1H), 2.63 (s, 3H), 2.53–2.38 (m, 1H), 2.11–1.90 (m, 3H).^13^C{^1^H}NMR (75.48 MHz, CDCl_3_) δ 163.0, 147.3, 144.8, 129.5, 129.0, 127.4, 125.7, 123.6, 118.6, 82.0, 69.1, 33.2, 25.9, 18.8.

**2-(2-hydroperoxytetrahydrofuran-2-yl)-4-methylquinoline 3aa’**. ^13^C{^1^H}NMR (75.48 MHz, CDCl_3_) δ 159.4, 146.0, 145.7, 128.8, 128.7, 127.2, 126.1, 123.4, 119.7, 113.3, 69.5, 36.8, 24.8, 19.0. HR-MS (ESI): *m*/*z* = 246.1125, calcd. for C_14_H_15_NO_3_+H^+^: 246.1123.

***Anti*-4-methyl-2-(5-methyltetrahydrofuran-2-yl)quinoline 3ab** was isolated using column chromatography (Petroleum ether/EtOAc = 2/1) as a colorless liquid (66 mg, 29%). ^1^H NMR (300 MHz, Chloroform-*d*) δ 8.07–8.02 (m, 1H), 7.98–7.93 (m, 1H), 7.66 (ddd, *J* = 8.4, 6.8, 1.5 Hz, 1H), 7.50 (ddd, *J* = 8.2, 6.8, 1.3 Hz, 1H), 7.46 (s, 1H), 5.26 (t, *J* = 7.1 Hz, 1H), 4.51–4.33 (m, 1H), 2.70 (s, 3H), 2.63–2.49 (m, 1H), 2.24–2.02 (m, 2H), 1.75–1.59 (m, 1H), 1.36 (d, *J* = 6.1 Hz, 3H). ^13^C{^1^H}NMR (75.48 MHz, CDCl_3_) δ 163.6, 147.3, 145.1, 129.6, 129.2, 127.5, 125.9, 123.8, 118.6, 81.8, 76.7, 34.1, 34.0, 21.5, 19.0; FTIR (KBr): ν_max_ = 2968, 2928, 2869, 1602, 1509, 1447, 1379, 1311, 1225, 1181, 1074, 910, 883, 760 cm^−1^. HR-MS (ESI): *m*/*z* = 228.1389, calcd. for C_15_H_17_NO+H^+^: 228.1383.

***Syn*-4-methyl-2-(5-methyltetrahydrofuran-2-yl)quinoline 3ab’** was isolated using column chromatography (Petroleum ether/EtOAc = 2/1) as a colorless liquid (59 mg, 26%). ^1^H NMR (300 MHz, Chloroform-*d*) δ 8.08–8.03 (m, 1H), 7.97 (dd, *J* = 8.4, 1.5 Hz, 1H), 7.68 (ddd, *J* = 8.4, 6.9, 1.5 Hz, 1H), 7.58–7.46 (m, 2H), 5.13 (dd, *J* = 7.6, 6.5 Hz, 1H), 4.33–4.21 (m, 1H), 2.72 (d, *J* = 0.7 Hz, 3H), 2.60–2.42 (m, 1H), 2.21–1.99 (m, 2H), 1.69–1.50 (m, 1H), 1.44 (d, *J* = 6.1 Hz, 3H).^13^C{^1^H}NMR (75.48 MHz, CDCl_3_) δ 163.3, 147.3, 145.2, 129.6, 129.3, 127.6, 126.0, 123.8, 118.8, 82.5, 76.9, 33.5, 33.2, 21.4, 19.1; FTIR (KBr): ν_max_ = 2970, 2928, 2870, 1736, 1602, 1563, 1509, 1447, 1380, 1090, 1032, 913, 882, 760 cm^−1^. HR-MS (ESI): *m*/*z* = 228.1388, calcd. for C_15_H_17_NO+H^+^: 228.1383.

**4-methyl-2-(2-methyltetrahydrofuran-2-yl)quinoline 3ab’** was isolated using column chromatography (Petroleum ether/EtOAc = 2/1) as a colorless liquid (28 mg, 12%). ^1^H NMR (300 MHz, Chloroform-*d*) δ 8.07 (d, *J* = 8.4, 1H), 7.99–7.94 (m, 1H), 7.67 (ddd, *J* = 8.4, 6.8, 1.5 Hz, 1H), 7.62–7.60 (m, 1H), 7.51 (ddd, *J* = 8.3, 6.9, 1.3 Hz, 1H), 4.13–4.02 (m, 1H), 3.95–3.83 (m, 1H), 2.88–2.75 (m, 1H), 2.71 (d, *J* = 1.0 Hz, 3H), 2.14–1.95 (m, 2H), 1.89–1.74 (m, 1H), 1.65 (s, 3H). ^13^C{^1^H}NMR (75.48 MHz, CDCl_3_) δ 166.6, 147.5, 144.6, 129.9, 129.0, 127.2, 125.8, 123.7, 118.5, 86.2, 68.1, 37.7, 28.3, 26.1, 19.1; FTIR (KBr): ν_max_ = 2977, 2931, 1600, 1447, 1383, 1363, 1196, 1101, 1033, 761 cm^−1^. HR-MS (ESI): *m*/*z* = 228.1380, calcd. for C_15_H_17_NO+H^+^: 228.1283.

**2-(1,3-dioxolan-2-yl)-4-methylquinoline 3ac** [65] was isolated using column chromatography (Petroleum ether/EtOAc = 2/1) as a colorless liquid (54 mg, 25%). ^1^H NMR (500 MHz, Chloroform-*d*) δ 8.16 (d, *J* = 8.4 Hz, 1H), 7.98 (d, *J* = 8.3 Hz, 1H), 7.74–7.67 (m, 1H), 7.59–7.53 (m, 1H), 7.49 (s, 1H), 5.95 (s, 1H), 4.27–4.19 (m, 2H), 4.16–4.08 (m, 2H), 2.71 (s, 3H). ^13^C{^1^H}NMR (75.48 MHz, CDCl_3_) δ 156.7, 147.2, 145.7, 130.2, 129.5, 128.4, 126.9, 123.8, 118.7, 104.3, 65.8, 19.0.

**2-(1,3-dioxolan-4-yl)-4-methylquinoline 3ac’** [65] was isolated using column chromatography (Petroleum ether/EtOAc = 2/1) as a colorless liquid (18 mg, 8%). ^1^H NMR (300 MHz, Chloroform-*d*) δ 8.05 (d, *J* = 8.4 Hz, 1H), 8.00 (dd, *J* = 8.4, 1.4 Hz, 1H), 7.71 (ddd, *J* = 8.4, 6.8, 1.4 Hz, 1H), 7.60–7.52 (m, 1H), 7.48 (s, 1H), 5.34 (s, 1H), 5.33–5.26 (m, 1H), 5.15 (s, 1H), 4.47–4.36 (m, 1H), 4.08 (dd, *J* = 8.3, 5.6 Hz, 1H), 2.73 (s, 3H). ^13^C{^1^H}NMR (75.48 MHz, CDCl_3_) δ 160.0, 147.1, 146.0, 129.7, 129.5, 127.8, 126.5, 123.9, 118.8, 96.4, 78.3, 71.1, 19.1. FTIR (KBr): ν_max_ = 2925, 2855, 16001, 1509, 1449, 1157, 1088, 1029, 936, 760 cm^−1^. HR-MS (ESI): *m*/*z* = 238.0841, calcd. for C_13_H_13_NO_2_+Na^+^: 238.0838.

**2-(1,4-dioxan-2-yl)-4-methylquinoline 3ad** [85] was isolated using column chromatography (Petroleum ether/EtOAc = 2/1) as white crystals (45 mg, 20%). Mp = 81–82 °C (lit. Mp = 82–83 °C [10.1039/C9OB02653C]). ^1^H NMR (300 MHz, Chloroform-*d*) δ 8.10 (d, *J* = 8.5 Hz, 1H), 7.98 (d, *J* = 8.7 Hz, 1H), 7.76–7.64 (m, 1H), 7.59–7.51 (m, 1H), 7.47 (s, 1H), 4.92 (dd, *J* = 10.3, 2.9 Hz, 1H), 4.25 (dd, *J* = 11.7, 2.9 Hz, 1H), 4.06–3.94 (m, 2H), 3.88–3.74 (m, 2H), 3.70–3.57 (m, 1H), 2.73 (s, 3H). ^13^C{^1^H}NMR (75.48 MHz, CDCl_3_) δ 157.9, 147.4, 145.3, 129.9, 129.4, 127.7, 126.3, 123.8, 119.2, 78.9, 71.2, 67.2, 66.5, 18.9.

**4-methyl-2-(tetrahydro-2H-pyran-2-yl)quinoline 3ae** [85] was isolated using column chromatography (Petroleum ether/EtOAc = 2/1) as a colorless liquid (73 mg, 32%). ^1^H NMR (300 MHz, Chloroform-*d*) δ 8.06 (d, *J* = 8.4 Hz, 1H), 7.98–7.89 (m, 1H), 7.70–7.60 (m, 1H), 7.52–7.46 (m, 1H), 7.45 (s, 1H), 4.60 (dd, *J* = 11.0, 2.3 Hz, 1H), 4.25–4.15 (m, 1H), 3.75–3.60 (m, 1H), 2.68 (s, 3H), 2.16–2.04 (m, 1H), 2.03–1.88 (m, 1H), 1.83–1.66 (m, 2H), 1.66–1.51 (m, 2H). ^13^C{^1^H}NMR (75.48 MHz, CDCl_3_) δ 162.2, 147.2, 145.1, 129.7, 129.1, 127.6, 125.9, 123.7, 118.9, 81.6, 68.9, 32.8, 25.9, 23.8, 18.9.

**2-(1-ethoxyethyl)-4-methylquinoline 3af** [85] was isolated using column chromatography (CH_2_Cl_2_/EtOAc = 20/1) as a colorless liquid (77 mg, 36%). ^1^H NMR (300 MHz, Chloroform-*d*) δ 8.07 (d, *J* = 8.4 Hz, 1H), 7.97 (d, *J* = 8.3 Hz, 1H), 7.68 (t, *J* = 8.4 Hz, 1H), 7.57–7.48 (m, 1H), 7.44 (s, 1H), 4.69 (q, *J* = 6.6 Hz, 1H), 3.57– 3.45 (m, 1H), 3.47–3.34 (m, 1H), 2.72 (s, 3H), 1.53 (d, *J* = 6.6 Hz, 3H), 1.22 (t, *J* = 7.1 Hz, 3H).^13^C{^1^H}NMR (75.48 MHz, CDCl_3_) δ 164.1, 147.2, 145.5, 129.6, 129.3, 127.8, 126.1, 123.8, 118.4, 79.7, 64.8, 22.7, 19.1, 15.6

**2-methoxy-4-(tetrahydrofuran-2-yl)quinoline 3ba** was isolated using column chromatography (EtOAc/petroleum ether 1/2) as a colorless liquid (122 mg, 53%). ^1^H NMR (300 MHz, Chloroform-*d*) δ 7.89 (d, *J* = 8.4 Hz, 1H), 7.81–7.72 (m, 1H), 7.68–7.54 (m, 1H), 7.44–7.31 (m, 1H), 7.07 (s, 1H), 5.52 (t, *J* = 6.9 Hz, 1H), 4.27–4.13 (m, 1H), 4.08 (s, 3H), 4.08–3.95 (m, 1H), 2.65–2.47 (m, 1H), 2.14–1.92 (m, 2H), 1.92–1.78 (m, 1H). ^13^C{^1^H}NMR (75.48 MHz, CDCl_3_) δ 162.8, 152.3, 147.1, 129.2, 128.1, 123.8, 123.3, 122.9, 108.5, 76.8, 69.1, 53.4, 33.7, 26.0. FTIR (KBr): ν_max_ = 2979, 2949, 1612, 1575, 1473, 1438, 1387, 1366, 1340, 1238, 1195, 1080, 1055, 1024, 761 cm^−1^. HR-MS (ESI): *m*/*z* = 230.1181, calcd. for C_14_H_15_NO_2_+H^+^: 230.1176

**2-chloro-4-(tetrahydrofuran-2-yl)quinoline 3ca** was isolated using column chromatography (Petroleum ether/EtOAc = 2/1) as a slightly yellow liquid (87 mg, 37%). ^1^H NMR (300.13 MHz, CDCl_3_) δ 8.04 (d, *J* = 8.5 Hz, 1H), 7.85 (dd, *J* = 8.4, 1.4 Hz, 1H), 7.71 (ddd, *J* = 8.4, 6.9, 1.4 Hz, 1H), 7.60–7.49 (m, 1H), 7.54 (s, 1H), 5.55 (t, *J* = 7.1 Hz, 1H), 4.22 (m, 1H), 4.02 (m, 1H), 2.70–2.55 (m, 1H), 2.13–1.95 (m, 2H), 1.94–1.76 (m, 1H).^13^C{^1^H}NMR (75.48 MHz, CDCl_3_) δ 153.1, 151.4, 148.1, 130.2, 129.5, 126.8, 124.5, 123.4, 117.9, 76.7, 69.2, 34.0, 26.1. FTIR (KBr): ν_max_ = 2965, 2928, 2871, 1586, 1560, 1506, 1292, 1264, 1145, 1099, 1081, 1041, 1021, 878, 855, 792, 763 cm^−1^. HR-MS (ESI): *m*/*z* = 234.0688, calcd. for C_13_H_12_ClNO+H^+^: 234.0680.

**2-methyl-4-(tetrahydrofuran-2-yl)quinoline 3ea** [91] was isolated using column chromatography (Petroleum ether/EtOAc = 2/1) as a colorless liquid (161 mg, 75%). ^1^H NMR (300 MHz, Chloroform-*d*) δ 8.03 (d, *J* = 8.3 Hz, 1H), 7.81 (d, *J* = 8.4 Hz, 1H), 7.69–7.57 (m, 1H), 7.50–7.41 (m, 1H), 7.42 (s, 1H), 5.53 (t, *J* = 7.1 Hz, 1H), 4.24–4.14 (m, 1H), 4.00 (q, *J* = 7.1 Hz, 1H), 2.65–2.47 (m, 1H), 2.12–1.89 (m, 2H), 1.86–1.72 (m, 1H). ^13^C{^1^H}NMR (75.48 MHz, CDCl_3_) δ 159.1, 149.4, 147.9, 129.4, 129.0, 125.5, 123.9, 123.0, 117.2, 76.8, 69.0, 33.9, 26.0, 25.6.

**2-(tetrahydrofuran-2-yl)quinoxaline 3fa** [66] was isolated using column chromatography (Petroleum ether/EtOAc = 2/1) as a colorless liquid (113 mg, 56%). ^1^H NMR (300.13 MHz, CDCl_3_) δ 9.02 (s, 1H), 8.12–8.07 (m, 1H), 8.07–8.01 (m, 1H), 7.76–7.69 (m, 2H), 5.21 (t, *J* = 7.0 Hz, 1H), 4.17 (q, *J* = 7.0 Hz, 1H), 4.05 (dd, *J* = 7.2 Hz, 1H), 2.57–2.46 (m, 1H), 2.21–2.11 (m, 1H), 2.11–2.00 (m, 2H). ^13^C{^1^H}NMR (75.48 MHz, CDCl_3_) δ 157.7, 143.6, 142.0, 141.7, 130.2, 129.6, 129.3, 129.2, 80.6, 69.5, 33.0, 26.1.

**2-(tetrahydrofuran-2-yl)pyrazine 3ga** was isolated using column chromatography (CH_2_Cl_2_/MeOH = 50/1) as a colorless liquid (59 mg, 40%).^1^H NMR (300 MHz, Chloroform-*d*) δ 8.68 (s, 1H), 8.52–8.36 (m, 2H), 5.01 (t, *J* = 6.4 Hz, 1H), 4.14–4.02 (m, 1H), 4.00–3.87 (m, 1H), 2.49–2.29 (m, 1H), 2.11–1.86 (m, 3H). ^13^C{^1^H}NMR (75.48 MHz, CDCl_3_) δ 158.2, 143.8, 143.5, 142.7, 79.8, 69.3, 32.9, 25.9. FTIR (KBr): ν_max_ = 3389, 2959, 2882, 1724, 1701, 1406, 1304, 1140, 1052, 1020 cm^−1^. HR-MS (ESI): *m*/*z* = 151.0873, calcd. for C_8_H_10_N_2_O+H^+^: 151.0866.

**2-methyl-3-(tetrahydrofuran-2-yl)pyrazine 3ha** was isolated using column chromatography (CH_2_Cl_2_/MeOH = 50/1) as a slightly yellow liquid (76 mg, 46%).^1^H NMR (300 MHz, Chloroform-*d*) δ 8.37 (d, *J* = 2.6 Hz, 1H), 8.34 (d, *J* = 2.6 Hz, 1H), 5.15 (t, *J* = 7.0 Hz, 1H), 4.13–4.04 (m, 1H), 3.99–3.89 (m, 1H), 2.63 (s, 3H), 2.31–2.19 (m, 2H), 2.17–1.96 (m, 2H). ^13^C{^1^H}NMR (75.48 MHz, CDCl_3_) δ 154.5, 152.5, 142.6, 141.5, 78.3, 69.2, 30.4, 26.3, 21.6. FTIR (KBr): ν_max_ = 3240, 3051, 2959, 2878, 1774, 1726, 1701, 1405, 1299, 1169, 1130, 1105, 1055, 988, 923, 857, 732 cm^−1^. HR-MS (ESI): *m*/*z* = 165.1023, calcd. for C_9_H_10_N_2_O+H^+^: 165.1022.

**Methyl 6-(tetrahydrofuran-2-yl)nicotinate 3ia** [91] was isolated using column chromatography (EtOAc/DCM = 1/20→1/5) as an orange liquid (119 mg, 57%). ^1^H NMR (300.13 MHz, CDCl_3_) δ 9.11 (d, *J* = 2.2 Hz, 1H), 8.25 (dd, *J* = 8.2, 2.2 Hz, 1H), 7.52 (d, *J* = 8.2 Hz, 1H), 5.11–4.95 (m, 1H), 4.32–3.74 (m, 5H), 2.57–2.33 (m, 1H), 2.09–1.81 (m, 3H). ^13^C{^1^H}NMR (75.48 MHz, CDCl_3_) δ 167.8, 165.9, 150.4, 137.9, 124.6, 119.4, 81.2, 69.3, 52.4, 33.2, 25.8.

**4-methyl-2-(tetrahydrofuran-2-yl)quinoline 1-oxide 3ja** [76] was isolated using column chromatography (Petroleum ether/EtOAc = 2/1) as a colorless liquid (78 mg, 34%). ^1^H NMR (300 MHz, Chloroform-*d*) δ 8.81–8.74 (m, 1H), 7.99–7.92 (m, 1H), 7.80–7.71 (m, 1H), 7.67–7.59 (m, 1H), 7.44 (s, 1H), 5.58 (t, *J* = 6.7 Hz, 1H), 4.17 (q, *J* = 6.9 Hz, 1H), 4.02 (q, *J* = 7.1 Hz, 1H), 2.90–2.76 (m, 1H), 2.68 (s, 3H), 2.13–1.98 (m, 1H), 1.99–1.82 (m, 2H). ^13^C{^1^H}NMR (75.48 MHz, CDCl_3_) δ 150.8, 141.1, 135.4, 130.3, 128.8, 128.0, 124.8, 119.9, 118.9, 76.1, 69.5, 31.2, 26.0, 18.6.

**1-(tetrahydrofuran-2-yl)isoquinoline 3ka** [91] was isolated using column chromatography (CH_2_Cl_2_/EtOAc from 5/1 to 5/2) as a colorless liquid (130 mg, 65%). ^1^H NMR (300.13 MHz, CDCl_3_) δ 8.50 (d, *J* = 5.8 Hz, 1H), 8.34 (d, *J* = 8.3 Hz, 1H), 7.82 (d, *J* = 8.1 Hz, 1H), 7.75–7.52 (m, 3H), 5.72 (t, *J* = 7.1 Hz, 1H), 4.20 (q, *J* = 7.3 Hz, 1H), 4.03 (q, *J* = 7.5 Hz, 1H), 2.60–2.32 (m, 2H), 2.27–2.01 (m, 2H).^13^C{^1^H}NMR (75.48 MHz, CDCl_3_) δ 159.7, 141.4, 136.7, 130.1, 127.5, 127.3, 126.7, 125.5, 120.7, 79.2, 69.1, 30.9, 26.3.

**2-(4-bromophenyl)imidazo [1,2-a]pyridine-3-carboxylic acid 3la’** [106] was isolated using column chromatography (CH_2_Cl_2_/EtOAc = 5/2) as slightly yellow crystals (82 mg, 30%). ^1^H NMR (300 MHz, Chloroform-*d*) δ 9.27 (bs, 1H, NH), 8.41 (d, *J* = 8.4 Hz, 1H), 8.28–8.19 (m, 1H), 7.84 (d, *J* = 8.5 Hz, 2H), 7.81–7.74 (m, 1H), 7.62 (d, *J* = 8.5 Hz, 2H), 7.09 (dd, *J* = 7.3, 4.9 Hz, 1H). ^13^C{^1^H}NMR (75.48 MHz, CDCl_3_) δ 165.1, 151.5, 147.1, 139.3, 133.1, 132.2, 129.2, 127.4, 120.2, 114.8.

## 4. Conclusions

In this work, a new visible-light active heterogeneous photocatalyst system based on industrially available and non-toxic TiO_2_ and NHPI was proposed for the cross-dehydrogenative C–C coupling of electron-deficient N-heterocycles with ethers. In this photocatalytic system, phthalimide-*N*-oxyl radicals photogenerated on the surface of titanium oxide become active mediators of the reaction, which leads to 1) an increase in efficiency due to the homogeneous organocatalytic process in solution and 2) allows the selective cleavage of the weak CH bonds. We have proposed a new mild method for the generation of C-centered radicals from non-activated esters for the Minisci reaction. Despite the fact that acidic additives are frequently used in Minisci-type reactions, the addition of acid was not necessary in our procedure in the case of several substrates. Optimal conditions were chosen for the Minisci reaction between π-deficient pyridine, quinoline, pyrazine, and quinoxaline heteroarenes with non-activated ethers.

## Data Availability

Not applicable.

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
