# Peer review of "Heterogeneous Photocatalysis as a Potent Tool for Organic Synthesis: Cross-Dehydrogenative C–C Coupling of N-Heterocycles with Ethers Employing TiO2/N-Hydroxyphthalimide System under Visible Light"

_molecules, 2023, doi:10.3390/molecules28030934_

Round 1

Reviewer 1 Report

The manuscript of Igor B. Krylov and Alexander O. Terent’ev entitled “Heterogeneous photocatalysis as a potent tool for organic synthesis: cross-dehydrogenative C–C coupling of N-heterocycles with ethers employing TiO2/N-hydroxyphthalimide system under visible light” describes a new variant of Minisci-type cross-dehydrogenative coupling between electron-deficient N-heterocycles and ethers employing TBHP as oxidant and NHPI/TiO2 photocatalytic system. Although many different catalytic systems were proposed previously for Minisci-type reactions, it is an important synthetic method in the chemistry of N-heterocycles which attracts continuous interest from the researchers. The present paper is an important expansion of the synthetic applicability of widely available and sustainable heterogeneous photocatalyst TiO2 in organic synthesis. The interaction of TiO2 with NHPI leads to the occurrence of activity under visible light, whereas no conversion to the target C-C coupling product was observed for TiO2 or NHPI used separately. Moreover, expected side processes of CH-oxygenation and C-O coupling between NHPI and ethers are avoided. I think this manuscript is suitable for publication after a minor revision:

The language should be checked. For example t-BuOO radicals are written as “tert-butyl peroxy” or “tert-butylperoxy”. One correct variant must be used.

The authors mentioned the detection of 3aa’ by NMR and HRMS: “The primary oxidation product was hydroperoxide 3aa’, which was detected in a mixture of oxidation products by 13С NMR and was confirmed by HRMS (See SI).”. However, the corresponding spectra are missing in SI. This should be corrected.

Structures should be deleted from the experimental part.

Author Response

Please, see the attached PDF file for point by point answers to all comments.

Reviewer 2 Report

This manuscript described the cross-dehydrogenative C–C coupling of N-heterocycles with ethers employing TiO2/N-hydroxyphthalimide system under visible light conditions. Various π-deficient pyridine, quinoline, pyrazine, and quinoxaline were functionalized using excess amounts of ethers. TBHP was the oxidant. Acid was not needed for some examples, while TFA was added to improve the yields in other cases.

Many catalytic systems in combination of different oxidants have been developed for this reaction. For example, 4,4ʹ-dibromobenzophenone as the photocatalyst with air as the oxidant (Tetrahedron Lett. 2022, 99, 153846), photocatalyst free with air as the oxidant (Org. Lett. 2021, 23, 6886−6890), photocatalyst free with H2O2 as the oxidant (New J. Chem., 2019, 43, 12533-12537), photocatalyst free with PIFA as the oxidant (Eur. J. Org. Chem. 2021, 411–421). Moreover, carbon nitride as a heterogeneous visible-light photocatalyst for this reaction for H2 production using air as the oxidant has also developed. In this work, a hybrid photocatalytic system was developed, which involves both heterogeneous catalyst (TiO2) and homogeneous organocatalyst (NHPI). Each of the catalysts is completely ineffective when used separately under visible light in this transformation. In my opinion, this catalytic system is no super than previous photocatalyst-free conditions with air as the green oxidant. My another complain is the low yields of the products. Only the model reaction gave 96% conversion and 89% yield. When the optimized conditions were applied to other substrates, the conversions were moderate while the yields were low to moderate.

The manuscript is well written and the supporting information is thorough and provides all the expected data for the new compounds. However, I feel that this work falls short on robustness, validity, and novelty, and thus do not recommend to be published in Molecules.   

Author Response

(The authors gave the same response as above.)

Round 2

Reviewer 2 Report

This manuscript can be accepted for publication in its present form.